# Isolated Valve Amyloid Deposition in Aortic Stenosis: Potential Clinical and Pathophysiological Relevance

**DOI:** 10.3390/ijms25021171

**Published:** 2024-01-18

**Authors:** Maddalena Conte, Paolo Poggio, Maria Monti, Laura Petraglia, Serena Cabaro, Dario Bruzzese, Giuseppe Comentale, Aurelio Caruso, Mariagabriella Grimaldi, Emilia Zampella, Annarita Gencarelli, Maria Rosaria Cervasio, Flora Cozzolino, Vittoria Monaco, Veronika Myasoedova, Vincenza Valerio, Adele Ferro, Luigi Insabato, Michele Bellino, Gennaro Galasso, Francesca Graziani, Pietro Pucci, Pietro Formisano, Emanuele Pilato, Alberto Cuocolo, Pasquale Perrone Filardi, Dario Leosco, Valentina Parisi

**Affiliations:** 1Department of Translational Medical Sciences, University of Naples Federico II, Via S. Pansini, 5, 80131 Naples, Italy; 2Casa di Cura San Michele, 81024 Caserta, Italy; carusoaurelio@gmail.com (A.C.);; 3Centro Cardiologico Monzino IRCCS, 20138 Milan, Italy; paolo.poggio@cardiologicomonzino.it (P.P.);; 4Dipartimento di Scienze Chimiche, University of Naples Federico II, 5, 80131 Naples, Italyflora.cozzolino@unina.it (F.C.); monacovi@ceinge.unina.it (V.M.);; 5CEINGE Biotecnologie Avanzate, Via Gaetano Salvatore 486, 80145 Naples, Italy; 6Department of Public Health, University of Naples Federico II, 5, 80131 Naples, Italy; 7Department of Advanced Biomedical Science, University of Naples Federico II, 5, 80131 Naples, Italycuocolo@unina.it (A.C.);; 8Institute of Biostructure and Bioimaging, CNR, 80145 Naples, Italy; 9Department of Medicine, Surgery and Dentistry, University of Salerno, Baronissi, 84081 Salerno, Italy; 10Department of Cardiovascular Medicine, Fondazione Policlinico Universitario A. Gemelli IRCCS, 00168 Rome, Italy; francesca.graziani@policlinicogemelli.it

**Keywords:** aortic stenosis, amyloidosis, inflammation, aortic valve interstitial cells, serum amyloid A

## Abstract

Amyloid deposition within stenotic aortic valves (AVs) also appears frequent in the absence of cardiac amyloidosis, but its clinical and pathophysiological relevance has not been investigated. We will elucidate the rate of isolated AV amyloid deposition and its potential clinical and pathophysiological significance in aortic stenosis (AS). In 130 patients without systemic and/or cardiac amyloidosis, we collected the explanted AVs during cardiac surgery: 57 patients with calcific AS and 73 patients with AV insufficiency (41 with AV sclerosis and 32 without, who were used as controls). Amyloid deposition was found in 21 AS valves (37%), 4 sclerotic AVs (10%), and none of the controls. Patients with and without isolated AV amyloid deposition had similar clinical and echocardiographic characteristics and survival rates. Isolated AV amyloid deposition was associated with higher degrees of AV fibrosis (*p* = 0.0082) and calcification (*p* < 0.0001). Immunohistochemistry analysis suggested serum amyloid A1 (SAA1), in addition to transthyretin (TTR), as the protein possibly involved in AV amyloid deposition. Circulating SAA1 levels were within the normal range in all groups, and no difference was observed in AS patients with and without AV amyloid deposition. In vitro, AV interstitial cells (VICs) were stimulated with interleukin (IL)-1β which induced increased SAA1-mRNA both in the control VICs (+6.4 ± 0.5, *p* = 0.02) and the AS VICs (+7.6 ± 0.5, *p* = 0.008). In conclusion, isolated AV amyloid deposition is frequent in the context of AS, but it does not appear to have potential clinical relevance. Conversely, amyloid deposition within AV leaflets, probably promoted by local inflammation, could play a role in AS pathophysiology.

## 1. Introduction

Calcific aortic stenosis (AS) is the most frequent valvular heart disease. Its prevalence increases with age and is the result of a complex and progressive process, lasting many years and promoted by mechanical stress and biological stimuli. From the early stages of the disease [1], the proinflammatory cytokines produced within the aortic valve (AV) [2,3] promote valve interstitial cell (VIC) differentiation and progressive AV leaflet fibrosis and calcification [4,5]. Aging is associated with a “low grade inflammatory status” that promotes and sustains several pathologic processes. This phenomenon, known as “inflammaging”, is characterized by increased levels of pro-inflammatory markers, including interleukin (IL)-6, IL-8, and C-reactive protein (CRP). It confers increased susceptibility to infections, metabolic abnormalities, cardiovascular disease, and impairments in physiological and physical function [6]. Increasing evidence suggests a high prevalence of myocardial amyloidosis in the context of AS, and the coexistence of the two diseases appears to be associated with a worse prognosis [7,8]. In a high percentage of AS patients (16%) the presence of transthyretin (TTR) amyloidosis within the myocardium has been reported [7]. TTR amyloidosis includes hereditary forms, caused by mutations in the TTR gene, and wild-type forms based on the absence of TTR mutations [9]. Amyloid deposition has also been described within the leaflets of surgically resected stenotic AVs [10], and isolated valvular amyloid deposition appears to be much more frequent than the dual disease (the valve and myocardium) [11]. High mechanical stress and increased local inflammation could promote the amyloidogenic process and justify the high prevalence of fibril accumulation within explanted stenotic valves. Singal et al. identified TTR in almost 57% of stenotic valves with amyloid deposits in patients without cardiac amyloidosis [11]. It is not well known which are the other proteins that can concur to amyloid fiber deposition within stenotic AVs. Furthermore, the clinical significance of isolated valve amyloid deposition and its potential role in AS pathophysiology remains unexplored. A deeper understanding of the pathophysiological mechanisms of calcific AS is highly desirable to identify new potential medical treatments able to halt, or at least slow, the AV degeneration process [12].

Aim of the Study. In the present manuscript, we explore the potential clinical significance of isolated AV amyloid deposition and its potential pathophysiological role in AS patients. We hypothesize that inflammation could be a trigger in isolated AV amyloid deposition. We aimed to test the potential involvement of serum amyloid A (SAA), an acute phase protein released in response to proinflammatory cytokines, in stenotic valve amyloid deposition. Within this scope, we enrolled patients undergoing cardiac surgery for different AV diseases, AS, and AV insufficiency in order to elucidate the possible role of “local” inflammation, typical of AS.

## 2. Results

The clinical and echocardiographic characteristics of AS patients are reported in Table 1 and Table 2. The AV sclerosis group was composed of 41 patients (32% females) with a mean age of 68 ± 7 years. The control group was composed of 32 patients (44% females) with a mean age of 57 ± 14 years. In all patients, there were no symptoms or signs of systemic amyloidosis.

The AVs explanted during cardiac surgery were evaluated for amyloid deposition at histology (Figure 1). Morphologically, amyloid deposition appeared diffuse and predominantly in the free part of the valve leaflet. In a few cases (n. 4), the deposition involved the annulus as well.

Light microscopy revealed amyloid deposits in 21 out of the 57 (37%) AS valves, 4 out of the 41 (10%) sclerotic AVs, and none of the AVs explanted from the controls. The presence of valve amyloid deposition was associated with both increased valve fibrosis and calcification (*p* = 0.0082 and *p* < 0.0001, respectively). In all patients with AV amyloid deposition, the bone scintigraphy was negative for myocardial uptake. To investigate the potential clinical significance of isolated AV amyloid deposition in AS, the population of AS patients was divided into two groups according to the presence of valve amyloid deposition. Within the two groups, there were no significant differences in clinical and echocardiographic parameters (Table 3).

We considered as a baseline in the survival analysis the clinical and echocardiographic evaluation performed during the pre-surgery period. No difference in patient survival was found between AS patients with and without AV amyloid deposition at 48 months follow-up (Figure 2). We separately evaluated data on early mortality (within 1 month from cardiac surgery), and no survival differences between AS patients with and without AV amyloid deposition were confirmed (*p* = 0.312).

***Immunohistochemistry.*** We analyzed during immunohistochemistry the 25 AVs with histological evidence of amyloid deposition, and the immunostaining was positive for SAA1 in 18 (72%) of the analyzed AVs and TTR in 15 (60%) of the analyzed AVs. In detail, 8 AVs (32%) were positive only for SAA1; 5 AVs (20%) were positive only for TTR; in 10 AVs (40%), the immunostaining was positive for both SAA1 and TTR; and in 2 AVs, the immunostaining was negative. In all of the samples, the immunohistochemical analysis resulted in being negative for kappa and lambda FLCs. To validate these results, immunohistochemistry analysis was also performed in a subgroup of AVs without amyloid deposition and the immunostainings were confirmed to be negative for both TTR and SAA1 in all examined samples.

***Circulating SAA levels in controls, AV sclerosis, and AS patients.*** Given that SAA amyloidosis is generally systemic and is the result of sustained abnormally high circulating levels of SAA, promoted by inflammation, we measured circulating SAA1 levels in the AS patients, AV sclerosis patients, and controls. The levels of circulating SAA1 were within the normal values [13] in the three groups (controls = 321.8 [191.1; 439.0] ng/mL; AV sclerosis = 270.3 [111.9; 453.6] ng/mL; AS = 207.1 [111.9; 346.3] ng/mL) and no significant differences were evidenced within the groups (controls vs. AV sclerosis *p* = 0.99; controls vs. AS *p* = 0.3309; AV sclerosis vs. AS *p* = 0.5041). Among the AS patients with and without AV amyloid deposition, no differences in circulating SAA1 levels were observed (266.1 [164.8; 381.1] ng/mL and 141.1 [86.59; 331.5] ng/mL, respectively, *p* = 0.0835).

***Aortic valve interstitial cells’ SAA1 production.*** Given the normality of circulating SAA1 values, we hypothesized that SAA1 could be produced within the AV leaflets. To test this hypothesis, we evaluated whether AV-VICs can produce SAA1 under inflammatory insults. We treated human-isolated VICs from six control valves as well as six stenotic ones with IL-1β or LPS (Figure 3A). In both control and stenotic AVs, VICs expressed SAA1 mRNA; however, VICs from stenotic valves expressed SAA1, in basal conditions, to a higher extent compared to the control VICs (+4.4 ± 0.3-fold, *p* = 0.02). In addition, VICs from the control and stenotic valves, under pro-inflammatory stimuli, showed a similar increase in SAA1 gene expression (IL1β: +6.4 ± 0.5-fold *p* = 0.01 and +7.6 ± 0.6-fold *p* = 0.008, respectively; LPS: +8.1 ± 0.3-fold *p* = 0.005 and +8.6 ± 0.3-fold *p* = 0.001, respectively), while no changes between the controls and stenotic VICs were found in TTR gene expression, either for the untreated condition or under pro-inflammatory stimuli (Figure 3B).

## 3. Discussion

The results of the present study indicate that, in severe AS patients, isolated AV amyloid deposition is frequent and has no relevant clinical significance, but it could have a role in AS pathophysiology. To be noted, we describe for the first time that SAA1 is present in the majority of AVs with amyloid deposits and could be the result of local protein production, promoted by local inflammatory stimuli rather than the result of an increment in circulating SAA levels promoted by systemic inflammation.

In AS patients screened for cardiac or systemic amyloidosis, the present study confirms a high rate of isolated AV amyloid deposition in the excised valves. Our data also suggest that amyloid deposits can be present at the AV sclerosis stage, while they were not present in the control AVs. To our knowledge, this is the first study looking at isolated valvular amyloid deposition in stenotic, sclerotic, and control AVs. Valvular sclerosis could justify the few cases of amyloid detection within insufficient valves described in previous studies [10]. Recent evidence reports that almost 60% of AVs with amyloid deposition were positive upon TTR immunostaining [11]. Our data are in line with this evidence but also suggest a high percentage (72%) of SAA1-positive immunostainings. SAA is an acute-phase protein, released in response to inflammation. SAA production is stimulated by proinflammatory cytokines, such as IL-6, IL-1, and transforming growth factor-𝛽 [14,15]. Although systemic SAA amyloidosis has been described as being associated with AS in the context of systemic inflammation [16], in our population, the SAA circulating levels were within the normal values, and no differences were observed in patients with and without AV amyloid deposition. This is the first study suggesting local, valvular, SAA-amyloid deposition in the absence of high circulating SAA levels. However, our data also suggest that SAA is not the only relevant amyloid protein, and further studies are required to provide a detailed characterization of amyloid fibrils within AVs.

We investigated the potential clinical and pathophysiological role of isolated valvular amyloid deposition in AS. In our population, we observed that among patients with and without AV amyloid deposition, there were no differences in the demographic, clinical, and echocardiographic parameters. Furthermore, although we observed a trend towards worse outcomes for AS patients with AV amyloid deposition, it did not reach statistical significance, suggesting that the presence of AV amyloid deposition had no impact on patient survival after AVR. These findings suggest that, in patients with severe calcific AS, there is no association between isolated AV amyloid deposition and clinical parameters. In this view, isolated AV amyloid deposition could constitute a completely different clinical scenario from cardiac amyloidosis which strongly impacts AS patient outcomes [17]. On the other side, our results raise new open questions on the possible role of amyloid deposition in AS pathophysiology. Previous studies have suggested that amyloid deposition within stenotic AVs promotes mineralization [18,19]. Accordingly, upon histological analysis, we found that amyloid within AVs is associated with higher degrees of leaflet fibrosis and calcium deposition. Also, the finding of amyloid deposits in the sclerotic AVs, but not in the control valves, further strengthens the hypothesis of a link between amyloid deposition and leaflet calcification. It is well known that AS is an atherosclerotic-like process promoted by local inflammation within the valve leaflets during the sclerotic phase of the disease [20]. The activation of inflammatory signaling pathways, including IL-1β, is involved in interstitial cell differentiation, valve matrix remodeling, and calcification [2,3,21]. Overall, the calcifying AV constitutes a site of high mechanical stress and sustained inflammation, two promoting factors of amyloidogenicity [22,23,24]. We investigated the hypothesis of local SAA increased production by assessing the VIC expression of SAA1 mRNA. VICs from stenotic valves expressed SAA1, in basal conditions, to a higher extent compared to VICs from the control valves. Furthermore, when we exposed AV-VICs to IL-1𝛽 or LPS in vitro, we observed a marked increase in SAA-mRNA production. Therefore, it is possible to hypothesize that local inflammation could trigger SAA protein overexpression. Interestingly, the same inflammatory stimuli did not upregulate TTR expression. Further studies are required to explore the role of inflammation in amyloid fiber formation in stenotic AVs.

## 4. Materials and Methods

### 4.1. Study Design and Patient Population

We prospectively enrolled 150 patients evaluated for AV replacement (AVR), including 75 patients with severe calcific AS and 75 patients with AV insufficiency. The exclusion criteria were patients undergoing transcatheter AVR, patients with rheumatic valve disease, suspected or confirmed systemic amyloidosis, or a chronic inflammatory/autoimmune disease, and patients with a history of infective endocarditis. The final study population was composed of 130 patients, 57 with AS and 73 with AV insufficiency. Out of the 73 patients with AV insufficiency, 41 patients had AV sclerosis at echocardiography whereas 32 patients, without AV sclerosis, were used as controls. As we excluded patients with suspected or confirmed systemic amyloidosis, no patients in the final study population showed clinical red flags for amyloidosis, such as carpal tunnel syndrome, lumbar canal stenosis, macroglossia, rupture of the long head of the bicep tendon, producing the ‘Popeye sign’.

### 4.2. Echocardiographic Study

All patients underwent a complete clinical and echocardiographic examination (Vivid E9, GE) before cardiac surgery. AV disease was classified into AS or AV insufficiency according to the ESC 2021 guidelines for the management of valvular heart disease [25]. In particular, a multiparametric evaluation was performed by evaluating valve morphology, color Doppler, and continuous and pulsed wave Doppler. The presence and severity of AS were determined by transvalvular velocity and gradients, AV area, flow status, and left ventricular systolic function. The presence and severity of AV insufficiency were determined by valve morphology, color flow regurgitant jet area, continuous wave Doppler signal of regurgitant jet, vena contracta width, pressure half-time, effective regurgitant orifice area, regurgitant volume, and left ventricle dilation. AV sclerosis was defined as AV calcification and thickening in the absence of stenosis. Speckle strain imaging was performed using an EchoPAC GE HealthCare workstation from transthoracic echocardiography in the 4-, 3-, and 2-chamber apical views. Regional longitudinal strain (LS) was determined in 17 segments of the left ventricle. Global LS was calculated as the average LS of these 17 segments. Relative apical LS was calculated as average apical LS/(average basal LS + average mid-LS).

### 4.3. Study Population Evaluation

The serum-free light chain (FLC) assay and immunofixation of serum and urine were performed in all patients to rule out light chain (AL) amyloidosis. We collected blood samples, and the centrifuged serum was stored at −80 °C and then screened for the concentration of SAA circulating levels through the use of an ELISA kit (Novus Biologicals, Littleton, CO, USA). AVR was performed through a standard median sternotomy and cardiopulmonary bypass. During cardiac surgery, the explanted AVs were collected and then analyzed through histology and immunohistochemistry analysis. In all patients with evidence of AV amyloid deposition at histology, we excluded the presence of TTR cardiac amyloidosis by 99mTcHydroxymethylene diphosphonate (99mTc-HMDP) scintigraphy [26]. Data on AS patients’ survival after AVR were collected in all patients through medical recordings and telephonic interviews.

### 4.4. Histology and Immunohistochemistry

Fresh tissue from valvular leaflets was transversely sectioned, fixed in 10% neutralized formalin, and subsequently embedded in paraffin. The valves rich in calcium deposits were pre-treated with a descaling solution (Bioptica ref. 05-03004Q) for 12 h. The FFPE sections (4 μm) were analyzed histopathologically and immunohistochemically. For the pathological analyses, the sections were stained with hematoxylin and eosin, with Mallory’s trichrome stain and Congo red stain. Under the optical microscope, fibrosis and calcium deposition were evaluated and graded as 0, 1+, 2+, and 3+ when absent or observed in <10%, 10–50%, and >50% of the sample, respectively. Congo red staining was used to assess the presence of amyloid deposition, considered positive when deposits that were initially visualized in bright light showed green birefringence in polarized light.

In the immunohistochemistry analysis, we evaluated the presence of the amyloidogenic proteins SAA, TTR, kappa, and lambda FLCs. The slides were incubated in antigen retrieval solution (Target Retrieval Solution Citrate [pH 6], Dako) at 110 °C in a bath for 10 min. After antigen retrieval, the slides were allowed to cool. The slides were rinsed with TBS, and the endogenous peroxidase was inactivated with 3% hydrogen peroxide. After protein block (BSA 5% in PBS 1×), the slides were incubated with primary antibodies at 4 °C overnight. The sections were incubated with an anti-mouse secondary IgG biotinylated antibody for 40 min. Immunoreactivity was visualized by means of avidin–biotin–peroxidase complex kit reagents (Novocastra, Newcastle, UK) as the chromogenic substrate. Finally, the sections were weakly counterstained with hematoxylin and mounted. The results of the immunohistochemistry analysis were based on the positive areas within the areas of amyloid deposition.

### 4.5. Aortic Valve Interstitial Cells’ Isolation and Culture

Isolation of VICs from 12 AVs, 6 stenotic AVs, and 6 control AVs was performed by implementing a method previously described [27]. The AV leaflets were incubated for 20 min at 37 °C in 2 mg/mL type II collagenase (Worthington Biochemical Corp. Lakewood, NJ, USA) in advanced Dulbecco’s modified Eagle’s medium (Ad DMEM, Life Technologies) containing 10% fetal bovine serum (FBSꞵ), 1% penicillin, 1% streptomycin, and 1% L-glutamine. Once the endothelial cells were removed, the leaflets were mechanically digested using a bistoury and then incubated for at least 4 h in 2 mg/mL type II collagenase at 37 °C. At the end of incubation, VICs were opportunely seeded. All of the experiments were performed on cultured cells between their second and fifth passage. IL-1β (PeproTech, London, UK) at 15 ng/mL or lipopolysaccharide (LPS) at 200 ng/mL was used as the inflammatory stimulus, and they were added to the cell medium every other day for 7 days.

### 4.6. Valve Interstitial Cells’ RNA Extraction and qPCR Assay

The extraction of RNA was performed from VICs using the Total RNA Purification Plus Kit (Norgen Biotek Corp, Thorold, ON, Canada). RNA was quantified using Nanodrop and used for two-step PCR amplification with a TaqMan Reverse Transcription Reagent kit (Life Technologies, Carlsbad, CA, USA). Total RNA (1 µg) was converted into cDNA. Real-time PCR (qPCR) was performed on ABI Prism 7900 HT (Applied Biosystems, Waltham, MA, USA), according to the manufacturer’s instructions, and analysis was performed using SDS2.4 software (Life Technologies, Carlsbad, CA, USA). Ribosomal Protein L32 (RPL32) was used as a housekeeping gene.

### 4.7. Statistical Analysis

Statistical analysis was conducted using the statistical platform R (vers. 4.0.1). Sample characteristics were reported using standard descriptive statistics. The mean ± standard deviation with the range was used in the case of numerical variables and absolute frequencies and percentages were used in the case of categorical factors. Numerical variables showing a highly skewed distribution were described using the median, interquartile range (25–75th percentile), and range. Accordingly, between-group comparisons were based on the *t*-test or the Mann–Whitney U test in the case of numerical variables and the chi-squared test or the Fisher’s exact test in the case of categorical outcomes. When comparing more than two groups, ANOVA or the Kruskal–Wallis test was used for omnibus statistics followed by the T-test or the Mann–Whitney U test. Adjusted differences among groups were estimated using median regression. Time-to-event outcomes were analyzed using the log-rank test and survival probabilities were estimated using the Kaplan–Meier method. In all analyses, two-sided *p*-values less than 0.05 were deemed as statistically significant, and no adjustment was made for multiplicity.

## 5. Conclusions

In conclusion, isolated AV amyloid deposition in AS can be considered as a distinct process from cardiac amyloidosis, with no relevant clinical significance at the end stage of the disease (severe AS) but with a potential pathophysiological role. The results of the present study generate the hypothesis that SAA1 overexpression within AV leaflets, triggered by local inflammation, could accelerate AS progression. Further studies are required to confirm this hypothesis and to investigate the potential role of other amyloidogenic proteins.

## 6. Study Limitations

Currently, immunohistochemistry is among the fastest methods available and does not require special equipment, but detectable amyloid proteins are limited by a set of antibodies. In addition, commercially available immunostaining antibodies do not have sufficient sensitivity or specificity, making it sometimes difficult to make an accurate typing of amyloidosis. Despite the methodological limitation of immunohistochemistry, we have made considerable effort to prevent us from misinterpreting the results, and the immunostainings were considered positive only when we observed the specific staining within the areas of amyloid deposition. Although our data suggest the possible involvement of SAA in AV amyloid deposition, our findings also suggest that SAA is not the only relevant amyloid protein. Further studies are required to provide a detailed characterization of amyloid fibrils within AVs.

We observed an association between amyloid deposition and AV leaflet calcification. However, the absence of amyloid fibrils in some stenotic AVs suggests that they are not necessary for the AV calcification process. Whether AV amyloid deposition has a role in AS progression should be verified by future studies.

We investigated the potential clinical significance of isolated AV amyloid deposition in AS by comparing clinical and echocardiographic parameters and survival at 48 months follow-up of AS patients with and without AV amyloid deposition. We did not find an association between isolated AV amyloid deposition and mortality. However, even in the absence of statistical significance, we observed a trend toward worse outcomes for AS patients with AV amyloid deposition, as shown in Figure 2. A longer follow-up and a larger sample size are required to confirm our findings.

## Figures and Tables

**Figure 1 ijms-25-01171-f001:**
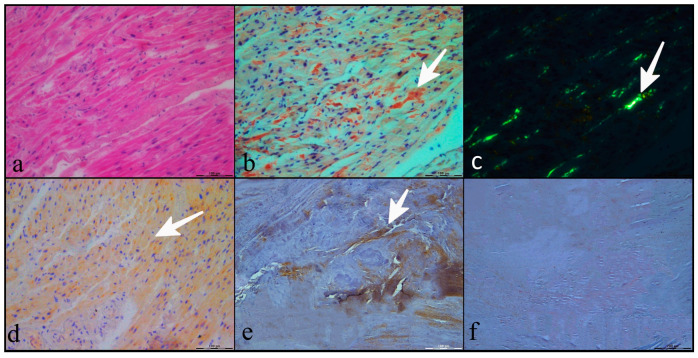
Aortic valve histology and immunohistochemistry. (**a**) Hematoxylin and eosin staining of a stenotic aortic valve; (**b**) evidence of amyloid deposits within the aortic valve through the use of Congo red staining; (**c**) apple-green birefringence at polarized light microscopy in the same area; (**d**) positive SAA1 staining during immunohistochemistry; (**e**) positive TTR staining during immunohistochemistry; (**f**) negative kappa/lambda staining during immunohistochemistry. The magnification of the figures is 100 μm.

**Figure 2 ijms-25-01171-f002:**
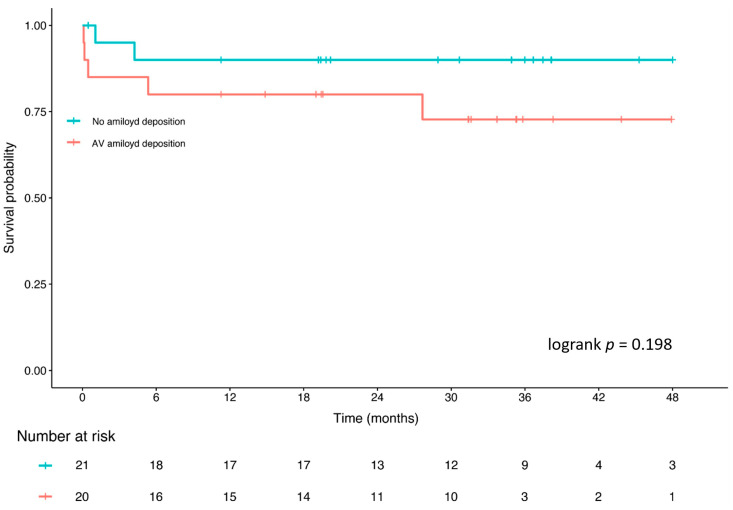
Survival probability of AS patients with and without isolated AV amyloid deposition. No survival differences were observed between patients with and without isolated AV amyloid deposition.

**Figure 3 ijms-25-01171-f003:**
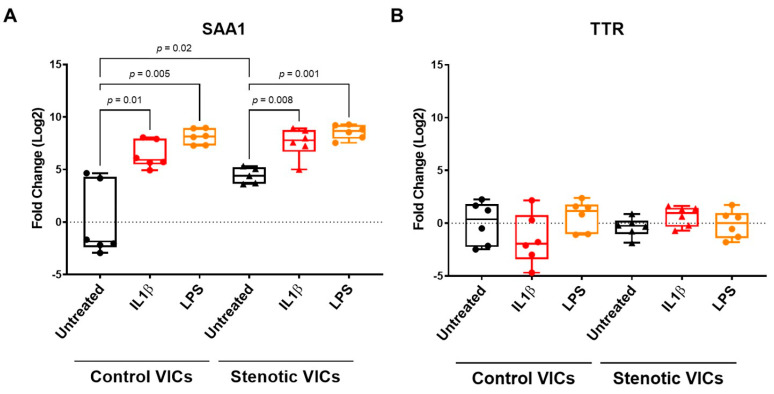
SAA1 and TTR gene expression in human-isolated valve interstitial cells. Box–Whisker plots representing SAA1 (**A**) and TTR (**B**) gene expression in valve interstitial cells (VICs) isolated from human control aortic valves (control VICs, n = 6) and human stenotic aortic valves (stenotic VICs, n = 6) under basal conditions (untreated) and after IL-1β or LPS stimulation. The fold changes were calculated against the untreated condition of the control VICs, which is the baseline for comparison. The statistical significance of the differences between the groups was tested by one-way ANOVA and post-hoc comparisons with Bonferroni’s multiple comparisons test.

**Table 1 ijms-25-01171-t001:** Clinical characteristics of AS patients.

Age (years)	68.9 ± 7.8 (50 to 84)
Gender female (n, %)	30 (72.6)
BMI (kg/m^2^)	27.3 ± 3.6 (16.4 to 35.3)
Hypertension (n, %)	51 (89.5)
Diabetes (n, %)	17 (29.8)
Smoking habit (n, %)	
Never	36 (63.2)
Current	15 (26.3)
Former	6 (10.5)
Dislipidemia (n, %)	29 (50.9)
Beta-blockers (n, %)	35 (64.8)
Calcium channel blockers (n, %)	10 (18.5)
Ace-inhibitors/sartans (n, %)	34 (61.8)
Statins (n, %)	28 (50.9)
Antiplatelets (n, %)	39 (70.9)

AS = aortic stenosis; BMI = body mass index.

**Table 2 ijms-25-01171-t002:** Echocardiographic characteristics of AS patients.

AS Severity (n, %)	
Mild	0 (0)
Moderate	3 (5.3)
Severe	54 (94.7)
Mixed valve disease	
Pure AS (n, %)	35 (61.4)
AS + mild AV insufficiency	17 (29.9)
AS + moderate AV insufficiency	5 (8.7)
LV ejection fraction (%)	62.4 ± 9.5 (37 to 80)
E/A	0.7 [0.6; 0.9] (0.3 to 3.8)
E/E’	13.1 ± 5.2 (3.7 to 29)
AV area (cm^2^)	0.8 ± 0.2 (0.4 to 1.2)
Mean gradient (mmHg)	47.7 [38.8; 55] (22.3 to 90)
LVEDD (mm)	46.9 ± 5.9 (38 to 58)
LVESD (mm)	28.1 ± 7 (12 to 46)
IVS thickness (mm)	11.9 ± 2.3 (6 to 16)
LV mass (g)	204 ± 61.4 (101 to 365.4)
LV mass index (g/m^2^)	116.3 ± 46.6 (1.1 to 261.8)
Granular sparkling (n, %)	0 (0)
Pericardial effusion (n, %)	4 (7)
GLS (%)	−16.1 ± 3.8
Relative apical LS	0.96 ± 0.04
Stroke volume index (ml/m^2^)	49.8 ± 13.8 (23.2 to 78)

AS = aortic stenosis; LV = left ventricle; AV = aortic valve; LVEDD = left ventricle end-diastolic diameter; LVESD = left ventricular end-systolic diameter; IVS = interventricular septum; GLS: global longitudinal strain; LS: longitudinal strain.

**Table 3 ijms-25-01171-t003:** Clinical and echocardiographic characteristics of AS patients (n. 57) with and without AV amyloid deposition.

	without AV Amyloid Deposition (n. 36)	with AV Amyloid Deposition (n. 21)	*p* Value
Age (years)	69.4 + −7.1 (52 to 84)	67.9 + −8.9 (50 to 82)	0.504
Gender female (n, %)	22 (61.1)	8 (38.1)	0.16
BMI (kg/m^2^)	27.3 + −3.6 (16.4 to 35.3)	27.3 + −3.8 (19.2 to 34.5)	0.998
Hypertension (n, %)	32 (88.9)	19 (90.5)	1
Diabetes (n, %)	10 (27.8)	7 (33.3)	0.887
Smoking habit (n, %)			0.838
Never	23 (63.9)	13 (61.9)	
Current	10 (27.8)	5 (23.8)	
Former	3 (8.3)	3 (14.3)	
Dislipidemia (n, %)	17 (47.2)	12 (57.1)	0.654
LV ejection fraction (%)	62.5 + −8.9 (40 to 75)	62.2 + −10.8 (37 to 80)	0.912
E/A	0.7 [0.6; 1] (0.3 to 3.8)	0.8 [0.6; 0.9] (0.5 to 1.1)	0.928
E/E’	13.4 + −5.8 (5 to 29)	12.5 + −4.2 (3.7 to 20)	0.544
AV area (cm^2^)	0.8 + −0.2 (0.4 to 1.2)	0.7 + −0.2 (0.5 to 1)	0.347
Mean gradient (mmHg)	45.5 [38; 54.5] (22.3 to 90)	52.4 [46; 56] (37 to 77.9)	0.201
LVEDD (mm)	46.9 + −6.1 (38 to 58)	46.9 + −5.7 (38 to 56)	0.998
LVESD (mm)	28.4 + −6.9 (18 to 46)	27.8 + −7.4 (12 to 39)	0.78
LV mass (g)	200.2 + −66.8 (101 to 365.4)	210 + −52.7 (137 to 304)	0.582
Stroke volume index (mL/m^2^)	49.6 [37; 64] (23.2 to 78)	46.7 [40.1; 50.5] (38.7 to 62)	0.541

AS = aortic stenosis; AV = aortic valve; BMI = body mass index; LV = left ventricle; LVEDD = left ventricle end-diastolic diameter; LVESD = left ventricular end-systolic diameter.

## Data Availability

The datasets used and/or analyzed during the current study are available from the corresponding author upon reasonable request.

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
