# Peer review of "Isolated Valve Amyloid Deposition in Aortic Stenosis: Potential Clinical and Pathophysiological Relevance"

_ijms, 2024, doi:10.3390/ijms25021171_

Round 1
Reviewer 1 Report
Comments and Suggestions for Authors
Minor Comments:
The hypothesis of the study should be more clearly formulated in the introduction, but not in conclusion.
The magnification in the histological figure is missing, please add.
Which houskeeping gene has been used in the Real-Time Experiments?
Author Response
- The hypothesis of the study should be more clearly formulated in the introduction, but not in conclusion.
We thank the reviewer for his/her comment. According to the reviewer suggestion, we added the hypothesis and the aim of our study in the introduction section (lines 73-80).
- The magnification in the histological figure is missing, please add.
We thank the reviewer for this comment. We added the missing information in the figure legend of Figure 1.
- Which houskeeping gene has been used in the Real-Time Experiments?
We thank the reviewer to point out the missing information. We used RPL32 as the housekeeping gene for the Real-Time Experiments. RPL32 is a ribosomal protein gene that has been shown to be stable and reliable for normalization in various tissues and cell types, including our cells. We have added this information in the Methods section, under the subsection “Valve Interstitial Cells RNA Extraction and qPCR Assay”.
Reviewer 2 Report
Comments and Suggestions for Authors
This manuscript investigates the presence of amyloid deposits in explanted aortic valves among patients without systemic or cardiac amyloidosis. The authors observe amyloid deposition associated with valve fibrosis and calcification, but no significant differences between patients with and without amyloid. Serum amyloid A protein or transthyretin was detected in most amyloid-containing valves. Interstitial cells from SAA-containing valves had a higher baseline expression of SAA than those where no SAA was detected.
This is an interesting manuscript that contains several thought-provoking observations. However, the strength of the evidence for several of the conclusions is weak and the manuscript needs to be altered to make this clearer. Rather than acquire additional data, which will be challenging, I suggest that the authors rewrite the manuscript in order to address these issues and tone down their conclusions.
Major issues
1. The study seems to lack the statistical power necessary to robustly reject the hypothesis that AV amyloid has clinical relevance. Notably, Figure 2 shows a trend towards worse outcomes for patients with amyloidosis, but there seem to be insufficient events to reach significance. Would a less strict endpoint be more useful?
2. Relatedly, It is difficult to keep track of which subset of patients is being compared at any point. Does Table 3 include all amyloid-containing valves, regardless of diagnosis? This should be made clear and total numbers should be included in the table.
3. There is only weak evidence that the SAA observed by IHC is actually AA amyloid. (The same is true for TTR, but TTR amyloid has been observed previously so this is a less surprising observation.)Given that the in vitro work is premised on the observation of AA amyloid, this lack of confirmation undermines the authors’ conclusions about the role of SAA. It would significantly strengthen the manuscript to show that the SAA has been proteolyzed or is insoluble, consistent with amyloid formation. Given that TTR was detected alongside SAA in 40% of samples and no amyloid protein was detected in 2 samples, I am not convinced that AA, and only AA, is the relevant amyloid protein. This does not necessarily the study, but the conclusions should be modified to reflect this possibility.
4. Local SAA production and subsequent local deposition of AA has not, to my knowledge, been described before. AA amyloidosis is rare, and only occurs in a fraction of individuals with chronic inflammation. It would be remarkable if the amyloid really is locally-produced AA, but as the manuscript stands, it seems to
5. It is not clear how the qPCR data is normalized, i.e., what is the fold-change relative to? Also, does a value of 5 on the “Fold change (log 2)” axis mean 5-fold or 32-fold?
6. Although the baseline SAA transcription is higher in cells from stenotic valves, the response of these cells to stimulus is similar to control cells. It is not clear what the relevance of the stimulus is. One might argue that the stenotic valve cells are actually less responsive to stimulus, which is not consistent with the authors’ hypothesis. Were the cells isolated from valves where SAA was detected? Also, 2/6 control cells have elevated SAA transcription, which undermines confidence in the result.
Minor issues
7. It would be helpful to define the relationship between AS, AV insufficiency and sclerosis. Are these conditions stages of a progression or distinct clinical entities? The abbreviations AS and AV sclerosis are potentially confusing for non-experts.
8. There are a number of grammatical quirks and errors that should be addressed. For example, “exalted” (line 60) is generally only used in a religious context – “increased” would be better.
9. The statistical parameters represented by the features of the boxplots in Figure 3 should be defined.
10. “Low grade inflammatory status” (line 52) should be defined and, ideally, referenced. Is SAA known to increase systemically with age?
11. The aims of the study (section 2.1) would be better in the introduction than in the methods.
Comments on the Quality of English Language
Another round of proofreading would be worthwhile.
Reviewer 3 Report
Comments and Suggestions for Authors
The authors prospectively investigated amyloid deposits in valvular leaflets obtained from 130 patients with aortic stenosis or aortic valve insufficiency. Amyloid deposition was found in 21 aortic stenosis valves, in 4 sclerotic aortic valves, and in none of the controls.
This study provides important insights into current knowledge on the pathogenesis and management of aortic stenosis. As novel therapeutic options appear one after another in the field of amyloidosis research, this manuscript will attract broad range of readers from basic scientists to physicians. The manuscript is well written, and I do not have any critical comments.
A suggestion to strengthen this manuscript is raised as follows:
1. Immunohistochemistry revealed that serum amyloid A1 and transthyretin were involved in amyloid deposition. Although most subjects with cardiac transthyretin amyloid deposition are considered to be wild-type transthyretin amyloidosis, patients with hereditary transthyretin amyloidosis may also be present (Cardiol Ther 2021; 10: 289-311). This issue should be incorporated in the introduction section, by citing this article.
Author Response
- A suggestion to strengthen this manuscript is raised as follows: Immunohistochemistry revealed that serum amyloid A1 and transthyretin were involved in amyloid deposition. Although most subjects with cardiac transthyretin amyloid deposition are considered to be wild-type transthyretin amyloidosis, patients with hereditary transthyretin amyloidosis may also be present (Cardiol Ther 2021; 10: 289-311). This issue should be incorporated in the introduction section, by citing this article
We thank the reviewer for his/her comment. We carefully read this relevant reference, and, as suggested, we implemented the introduction section (lines 60-61) and included this reference in the revised manuscript (n.9 of the revised manuscript)